# TP53 Status, Patient Sex, and the Immune Response as Determinants of Lung Cancer Patient Survival

**DOI:** 10.3390/cancers12061535

**Published:** 2020-06-11

**Authors:** Donald Freudenstein, Cassandra Litchfield, Franco Caramia, Gavin Wright, Benjamin J. Solomon, David Ball, Simon P. Keam, Paul Neeson, Ygal Haupt, Sue Haupt

**Affiliations:** 1Tumor Suppression Laboratory, Peter MacCallum Cancer Centre, 305 Grattan St, Melbourne, 3000, Australia; donald.freudenstein@gmail.com (D.F.); Cassie.Litchfield@petermac.org (C.L.); Franco.Caramia@petermac.org (F.C.); Simon.Keam@petermac.org (S.P.K.); Ygal.Haupt@petermac.org (Y.H.); 2Department of Surgery, St Vincent’s Hospital, The University of Melbourne, Fitzroy, VIC 3065, Australia; lung_surgeon@me.com; 3Department of Medical Oncology, Peter MacCallum Cancer Centre, Melbourne, VIC 3000, Australia; Ben.Solomon@petermac.org; 4Sir Peter MacCallum Department of Oncology, The University of Melbourne, Parkville, VIC 3010, Australia; David.Ball@petermac.org (D.B.); Paul.Neeson@petermac.org (P.N.); 5Department of Radiation Oncology Peter MacCallum Cancer Centre, 305 Grattan St, Melbourne, VIC 3000, Australia; 6Cancer Immunology Research, Peter MacCallum Cancer Centre, 305 Grattan St, Melbourne, VIC 3000, Australia; 7Department of Clinical Pathology, University of Melbourne, Parkville, VIC 3010, Australia; 8Department of Biochemistry and Molecular Biology, Monash University, Melbourne, VIC 3004, Australia

**Keywords:** *TP53*, lung cancer, sex disparity, LUAD, LUSC, immune signatures

## Abstract

Lung cancer poses the greatest cancer-related death risk and males have poorer outcomes than females, for unknown reasons. Patient sex is not a biological variable considered in lung cancer standard of care. Correlating patient genetics with outcomes is predicted to open avenues for improved management. Using a bioinformatics approach across non-small cell lung cancer (NSCLC) subtypes, we identified where patient sex, mutation of the major tumor suppressor gene, Tumour protein P53 (TP53), and immune signatures stratified outcomes in lung adenocarcinoma (LUAD) and lung squamous cell carcinoma (LUSC), among datasets of The Cancer Genome Atlas (TCGA). We exposed sex and TP53 gene mutations as prognostic for LUAD survival. Longest survival in LUAD occurred among females with wild-type (wt) TP53 genes, high levels of immune infiltration and enrichment for pathway signatures of Interferon Gamma (INF-γ), Tumour Necrosis Factor (TNF) and macrophages-monocytes. In contrast, poor survival in men with LUAD and wt TP53 genes corresponded with enrichment of Transforming Growth Factor Beta 1 (TGFB1, hereafter TGF-β) and wound healing signatures. In LUAD with wt TP5*3* genes, elevated gene expression of immune checkpoint CD274 (hereafter: PD-L1) and also protein 53 (p53) negative-regulators of the Mouse Double Minute (MDM)-family predict novel avenues for combined immunotherapies. LUSC is dominated by male smokers with TP53 gene mutations, while a minor population of TCGA LC patients with wt TP53 genes unexpectedly had the poorest survival, suggestive of a separate etiology. We conclude that advanced approaches to LUAD and LUSC therapy lie in the consideration of patient sex, TP53 gene mutation status and immune signatures.

## 1. Introduction

Lung cancer remains the greatest cause of cancer death worldwide [1], including the United States (US) [2]. Less than 50% of lung cancer patients survive one year following diagnosis, with only 18% remaining alive after five years (reviewed in [3]). Reducing deaths from lung cancer using rational, pathogenesis-directed therapies is a prime motivation for establishing the underlying processes causing lung cancer development and tumor progression.

Non-small cell lung cancer (NSCLC) constitutes ~85% of lung cancers, while ~15% are classified as small cell lung cancer (SCLC). NSCLC is further classified into subtypes which are dominated by lung adenocarcinoma (LUAD) and lung squamous cell carcinoma (LUSC), respectively (reviewed in [4]). Currently the cases of LUAD outnumber those of LUSC in the US by more than 2:1 [5]. Histopathology is the gold standard for diagnosis and subtype discrimination, with staging defined using computer topography (CT), positron emission tomography (PET), and magnetic resonance imaging (MRI) (reviewed in [6]). Deep learning algorithm analyses are demonstrating potential for improved diagnosis [7] and disease staging [8]. RNA profiling is being intensively pursued to identify biomarkers to more precisely discriminate these subtypes. The aim of detailed NSCLC characterization is to guide clinical management and appropriate treatment design, for example, where anti-angiogenic therapy is beneficial in LUAD and not in LUSC (reviewed in [9]).

LUSC is predominantly a result of carcinogenic smoke exposure from previous decades of inhalation from unfiltered cigarettes [10]. LUSC rates are declining [5] and this is largely due to decreased rates of smoking in men, as they were more prone to smoke unfiltered cigarettes than women [10]. In contrast, over the same period, counter-intuitively, the introduction of cigarette filter ventilation corresponds with higher rates of LUAD. One postulated explanation is that milder smoke composition prompts extended and deeper inhalation, with greater risk to more recessed distal lung cells that are more susceptible to adenocarcinoma [10]. Whether pollutants (referenced in [11]), associated with increased air pollution (including bush fire, and volcano smoke and ash) and additional altered smoking habits (vaping and nargila use) are impacting on the rising LUAD incidence remains to be established. 

Overall survival outcomes for NSCLC were found to be more favorable for females than males in US and Australian populations [12], consistent with earlier data adjusted for age, histology, and disease stage [13,14,15,16]. What drives this sex disparity in survival remains to be clearly established. The expectation that such factors are of therapeutic relevance is motivating their exploration.

We recently reported significant correlation between poor overall survival and mutation of the major tumor suppressor gene TP53 for the vast majority of US patients with spontaneous cancers of nonreproductive organs [17]. Of particular relevance to lung cancer, we calculated that TP53 mutation risk is higher in NSCLC for males than females, in the US population [17]. Whether high incidence of TP53 mutation in men correlates with poor survival for LUAD and LUSC individually is particularly relevant due to the established link between smoking and TP53 mutation [18]. 

The immune system is now a recognized determinant of cancer outcomes (reviewed in [19]) and links are emerging to p53 protein across many levels (reviewed in [20,21]; and we discriminate the gene as TP53 and its protein as p53, from this point). In lung epithelia, a p53/microRNA-34(miR-34)/PD-L1 axis is reported to dictate immune detection, with p53 dysfunction tied to elevated PD-L1 expression and immune evasion in LUAD [22]. Relevant to antigen presentation, the peculiar structural stability of each individual p53 protein mutation dictates the nature of its presentation to T cells, with greater instability eliciting higher immune response [23]. The TP53 mutation status of tumor cells can also profoundly influence the immune response in its surrounding microenvironment (reviewed in [24,25]), by regulating inflammation [26,27].

While links between p53, mutation of its encoding gene TP53, and immune responses are emerging, a corresponding influence of patient sex on outcome is yet to be ascertained. This is relevant, as distinct immune profiles are evident between females and males (reviewed in [28]). Motivated to investigate possible connections between TP53 mutation, patient sex, immunity, and disease outcomes in the two major NSCLC subtypes, we were prompted to explore for fate-determining engagements.

## 2. Results

### 2.1. Females with LUAD Survive Longer Than Their Male Counterparts 

Aware that female patients diagnosed with NSCLC were reported to outlive their male counterparts in Australia and the US in recent statistics [12], we were curious to examine whether this held for its main component subtypes, LUAD and LUSC, and whether this is connected to TP53 status. As LUAD and LUSC are distinct subtypes, we elected to analyze them separately. We initially focused on the more prevalent LUAD and subsequently LUSC. We examined LUAD patient overall survival in the US Surveillance, Epidemiology, and End Results Program (SEER) data National Cancer Institute (NCI), and in an Australian cohort (as described in Section 4.1 Materials and Methods). Our analyses revealed that female LUAD patients in both these Western populations survive longer than their male counterparts: In the US (Figure 1A, *p* < 0.0001; Appendix A, *p* < 0.0001; with 2016 being the last year of data collection) and Australia (Figure 1B, *p* < 0.001 for patients diagnosed 2013–2017). 

Relative LUAD incidence between the sexes in the US population from 2001-2016 was also extracted from SEER data [5] (Appendix A). It is notable that in both US and Australian whole population datasets, the latest incidence of LUAD among males and females was similar, although in each population the rates were slightly higher for males.

### 2.2. Female LUAD Patients with wt TP53 Status Have the Most Favorable Survival Outcomes 

Our recent findings demonstrate that for the vast majority of patients with nonreproductive cancers worse survival corresponds with TP53 mutation [17]. In this current study, we sought to test whether this was evident across the major subtypes of NSCLC. Again, LUAD analyses are presented first and LUSC later. For LUAD patients, our new findings in The Cancer Genome Atlas (TCGA) clearly demonstrated better overall survival in the context of wild-type (wt) TP53 (Figure 1C, *p* = 0.0016, according to numbers of LUAD patients listed in Table 1). As females had a better overall survival for LUAD (Figure 1A), we next stratified for both TP53 status and patient sex (Figure 1D, *p* < 0.001). Female LUAD patients with wt TP53 had the most extended overall survival compared with other LUAD patients (Figure 1D,E, *p* = 0.0034). In contrast, male survival did not significantly differ between patients with wt or mutant TP53 (Figure 1F, *p* = 0.11).

### 2.3. Immune Infiltrate Is More Abundant in wt TP53 LUAD of Females than Males

Propensity scores are frequently applied to TCGA data to ensure that underlying causes of outcome are not masked by confounding factors (e.g., [29]). Adjustment for tumor purity is one common correction undertaken. In our study of the TCGA samples, we adopted Consensus measurement of Purity Estimations (CPE, refer to Materials and Methods Section 4.2, reviewed in [30]) to estimate tumor purity (Appendix A). Tumor infiltrate is the component of the biopsy that is not tumor (calculated as one minus the ‘tumor purity’ proportion for each LUAD sample). We identified that the levels of infiltrate varied widely across the LUAD cohorts (Figure 2A). 

Importantly, tumor infiltrate is comprised of two major components: Immune infiltrate and stroma (reviewed in [30]). With heightened interest in the influence of tumor infiltrating lymphocytes on patient outcomes (e.g., in cancer of the colon [31] and the breast [32]), we chose to investigate the nature of this infiltrate in LUAD samples. Immune infiltrate was quantified from the ‘leukocytes fraction’ data based on DNA methylation arrays ([33], refer to Section 4. Material and Methods, Appendix A). The significant Pearson correlation coefficient (0.85, *p* < 0.05, Figure 2B) between the immune infiltrate and the tumor infiltrate indicates that the dominant no tumor cells were of hematological origin and were not stroma. Specifically, the calculated Pearson coefficient value close to 1 indicates a significant positive linear correlation, using the Spearman’s test of significance. We considered that rather than low tumor purity/high tumor infiltrate being a ‘problem’ (as in other studies which discard samples of low purity, e.g., [34]), we postulated that the nature and quantity of immune infiltrate may be prognostic in female and male LUAD patients. 

As longevity is more prolonged for females than males with LUAD (Figure 1A) and this is most evident for female wt TP53 cases (Figure 1D,E), we examined the proportion of immune infiltrate across the sexes according to TP53 status (wt or mutant). Notably, female LUAD with wt TP53 had significantly higher levels of immune infiltrate than their male counterparts (Figure 2C, *p* < 0.01; while no significant difference was evident in LUAD with mutant TP53 of either sex (Figure 2D, *p* = 0.56). We chose to compare the composition of these infiltrates across the cohorts comprising LUAD.

### 2.4. The Composition of the Immune Infiltrate Is Distinct between LUAD in Males and Females 

The composition of the immune infiltrate was characterized using the approach of Thorsson et al., 2018 [33], to discriminate five immune signature gene sets: Interferon-gamma response (IFN-γ), lymphocytes’ infiltration [35], macrophage-monocyte regulation [36], TGF-beta (TGF-β) response [37], and wound healing [38] (refer to Materials and Methods Section 4.1). To examine for differences in enrichment of these immune signatures among differentially expressed genes between males and females in LUAD, Gene Set Enrichment Analysis (GSEA, refer to Materials and Methods Section 4.6) was applied. Analyses were performed separately for patients with wt TP53 and mutant TP53. 

Comparison between the sexes for wt TP53 LUAD identified enrichment in three of the five immune signatures in females: macrophages and monocytes (adjusted *p* = 0.00089, normalized enrichment score (NES) = 2.28955), lymphocyte infiltration (adjusted *p* = 0.00423, NES = 1.70500), and interferon-γ ((IFN- γ) adjusted *p* = 0.00931, NES = 1.65516, Figure 2E, Appendix A). In contrast, in the mutant TP53 LUAD context, the findings were completely different, with enrichment in females only of the gene signature for the TGF-β pathway and in males only of the wound healing signature (Figure 2F, Appendix A). As these immune signatures are constructed from networks of genes (Appendix A), the next undertaking was to identify which of the components genes in these pathways exhibited significant sex-disparity in their expression. As extended survival was most significant among LUAD patients with wt TP53 and cancers were significant, we focused on GSEA analysis of the constituent immune gene networks of this group.

### 2.5. The wt TP53 LUAD in Females Is Enriched for Antigen Processing Genes Corresponding with Superior Survival Compared with Their Male Counterparts

Macrophages chemically stimulated in vitro have been categorized by their responses to be either M1 pro-inflammatory or M2 anti-inflammatory/pro-resolving (reviewed in [39]). This system of classification has limitations in the in vivo setting and suggestions are being proposed for its revision. In the absence of a more precise descriptor at present, we adopted this M1/M2 system, but are aware that it is not perfect (reviewed in [40]. Distinct gene signatures correspond to the M1 and M2 populations (Appendix A). The M1 macrophage gene signature (as defined in [41]) is positively enriched in wt TP53 LUAD in females, compared with males, using GSEA analysis (Figure 3A, Appendix A, adjusted *p* = 0.00068, NES = 2.03845). In contrast, the M2 signature trended toward enrichment in wt TP53 LUAD in males, but not to a level of statistical significance (Figure 3A, Appendix A, adjusted *p* = 0.48778, NES = −1.091991). To examine this further, we looked at the differential expression of the individual genes comprising the M1 macrophage signature (Appendix A). Among the M1 signature genes, *SUSD3* stands out for its significant, high expression in females, (Figure 3B, Appendix A, adjusted *p* = 0.01381), in addition to X-linked gene TSIX (Appendix A).

The immunomodulator genes (Appendix A) Major Histocompatibility Complex, Class II DR Beta 1 (HLA-DRB1), Interferon Alpha and Beta Receptor Subunit 2 (IFNAR2) and Tumour Necrosis Factor (TNF) are more highly expressed in wt TP53 LUAD in females, compared with their male counterparts, according to differential gene expression analysis comparison (Figure 3C, Appendix A, TNF adjusted *p* = 0.021371, HLA-DRB1 adjusted *p* = 0.046451, and IFNAR2 adjusted *p* = 0.020318). The tumor inflammation signature (TIS) is comprised of 18 genes (Appendix A) and is used to indicate the level of suppression of adaptive immunity in tumors and differentiate response to the checkpoint anti-PD-1 treatment (where PD-1 is encoded in the Programmed Cell Death 1 [PDCD1] gene) [42]. HLA-DRB1 also stands out among the genes of the TIS as highly expressed in females compared with males (Figure 3D, Appendix A, adjusted *p* = 0.046451). Of note, the expression of the remaining majority of the genes (16/18) from the TIS signature is higher in females than the males, although not crossing the threshold of significance (Figure 3D, Appendix A). It is pertinent to note that the single gene trending higher in males is CD276, which is an important immune checkpoint molecule that inhibits T cell function (reviewed in [43]; although its expression also does not cross the significance threshold). Also, for the lymphocyte infiltration immune signature, the majority of genes (15/16) are expressed at higher levels in females, but not to a level of significance (Appendix A).

Examining the differentially expressed genes between wt TP53 males and wt TP53 females against the antigen processing pathway from the Kyoto Encyclopedia of Genes and Genomes (KEGG) (refer to Materials and Methods Section 4.8) revealed higher levels of expression for genes of this central immune pathway in females (Figure 4A). High levels of expression in females are indicated by red outlining. 

Overall, these analyses are consistent with robust cancer immunity, corresponding with high expression of antigen presentation genes, associated with high levels of lymphocyte infiltration and macrophage signatures driving extended survival among wt TP53 LUAD in female patients.

### 2.6. A Novel Immune Gene Expression Signature Corresponds with Extended Overall Survival Predominately among wt TP53 LUAD in Females 

The overall survival of LUAD patients was next analyzed according to high and low levels of expression of the five immune gene sets introduced in Section 2.4. Lymphocytes’ infiltration [35], macrophage-monocytes [36], interferon-gamma response (IFN-γ), TGF-beta response (TGF-β) [37] and wound healing [38] (Table 2, Appendix A, respectively). Significantly better survival corresponds with high expression levels of lymphocytes infiltration genes (Appendix A, *p* = 0.031). Significantly worse survival was evident among LUAD patients with high expression levels of the TGF-β gene set (Appendix A, *p* = 0.02) and wound healing genes (Appendix A, *p* ≤ 0.0001).

We then examined overall survival of LUAD patients in the TCGA cohort stratifying patients by sex, TP53 status, and expression levels of genes in the three pathways that were of greatest significance in females: for the M1 macrophage signature gene Sushi Domain Containing 3 (SUSD3); the immunomodulator genes Major Histocompatibility Complex Class II DR Beta 1 (HLA-DRB1) and Tumour Necrosis Factor (TNF) (Appendix A: SUSD3 *p* = 0.0015, HLA-DRB1 *p* = 0.0018, and TNF *p* = 0.014). Note that and the significant gene HLA-DRB1 is also part of the Tumour Inflammation Signature (TIS). 

The significant correspondence of the expression of these genes to patient outcomes individually led us to define a combined, fate-determining immune signature in LUAD, specifically comprised of: SUSD3, HLA-DRB1, and TNF (Figure 4B). 

### 2.7. LUSC Is Dominated by Males with Mutant TP53

As we introduced, because LUSC is a distinct subtype from LUAD, we elected to analyze it independently, while using the same analytical approaches. We examined the survival of females and males with LUSC in Western populations, with adjustment for confounders (described in Section 4.1). Female survival in the US SEER data was not statistically different from males (Figure 5A, *p* = 0.18; and Appendix A, *p* = 0.08; with 2016 being the last year of collection) and this was replicated in the Australian population (collected between 2013–2017, Figure 5B, *p* = 0.275). Relative LUSC incidence between the sexes in the US population for the extended dates of 2001–2016 was also extracted from SEER data [5] (Appendix A) and revealed a similar trend (*p* = 0.08). Notably, male incidence with LUSC exceeded that of females from both the population of the US (males: 1825, females: 1080, Figure 5A; males: 6177, females: 3515, Appendix A) and Australia (males: 1730, females: 777, Figure 5B). This is in direct contrast to the near equivalent incidence between the sexes for the LUAD datasets (males: 5475, females: 5430 in the US, Figure 1A and Appendix A, males: 17,207, females: 16,297 and in Australia, males: 3044, females: 2720, Figure 1B). 

An overwhelming 96% of the LUSC patients in our TCGA analyses had identified smoking history (Appendix A) as relevant to causation. Among these, a striking >86% of the females and males combined had mutant TP53 (Figure 5C; Table 1), consistent with its reported association with smoking [44]. This was distinct from the near equal ratio of wt to mutant TP53 content in the LUAD TCGA dataset (Figure 1C, Table 1), although it is noted that TCGA does not claim representative population sampling. Unexpectedly, patients with wt TP53 LUSC had significantly reduced survival compared with their mutant TP53 counterparts (Figure 5D, *p* < 0.0001). This was the opposite outcome from LUAD, as we show here (Figure 1), and also to the overall survival in most nonreproductive cancers [17], where TP53 mutation corresponds with poorest survival. Reduced survival among LUSC patients with wt TP5*3* is a trend for both females and males (Figure 5E, p = 0.07 and Figure 5F, *p* = 0.17, respectively, with comparably fewer patients with wt TP53 apparently limiting the measurable significance). 

### 2.8. LUSC Tumors Are Infiltrated by Both Immune Cells and Other Cells

The tumor infiltrate of TCGA LUSC samples was heterogeneous (Figure 6A), with lower correlation with the immune infiltrate (Figure 6B, Pearson correlation = 0.67) than measured for LUAD (Figure 2B, Pearson correlation = 0.85). Significantly higher levels of immune infiltrate were measured in LUSC from females than from males, for both wt and mutant TP53 (Figure 6C, *p* = 0.02; Figure 6D, *p* < 0.01) in contrast to LUAD, where significance was only seen for wt TP53 cancers (Figure 2C,D). Unfortunately, due to the low numbers of wt TP53 LUSC patients in the TCGA data, resulting in a lack of statistical power, additional immune pathway analyses for this group are not presented. 

As mutant TP53 dominates TCGA LUSC datasets, this population was the focus of the subsequent analyses. The immune signatures in mutant TP53 LUSC are discriminated by patient sex, with females enriched in the signatures for IFN-γ response (Figure 6E, Appendix A, adjusted *p* = 0.00852, NES = 1.65420), lymphocyte infiltration (adjusted *p* = 0.00277, NES = 1.72073), and macrophage-monocytes (adjusted *p* = 0.00086, NES = 2.25607). Unexpectedly, this was remarkably similar to the findings for female LUAD wt TP53 (Figure 2E) and in complete contrast to LUAD with mutant TP53 (Figure 2F). TGF-β response and wound healing were more enriched in LUSC with mutant TP53 in males than females, but not to a significant level (Figure 6E, Appendix A), as also observed for wt TP53 LUAD in men (Figure 2E). Further, the lymphocyte infiltrate immune signature was enriched in females compared with males with mutant TP53, but not to a significant level (Appendix A). Immune signatures in LUSC with mutant TP53 did not exhibit any prognostic value for overall survival (Table 3), in contrast to those in LUAD (Table 2).

### 2.9. Survival Outcomes Were Not Predicted by Immune Genes in Mutant TP53 LUSC

The macrophage M1 signature was significantly enriched in mutant TP53 LUSC in females (Figure 6F, Appendix A, adjusted *p* = 0.00025, NES = 2.06806), while the M2 signature was dominant in males (adjusted *p* = 0.000, NES = −2.14073). Three genes were more significantly expressed in the M1 signature in females: Pleckstrin Homology Domain Containing O1 (PLEKHO1), Guanylate Binding Protein 4 (GBP4), and C-C Motif Chemokine receptor 7 (CCR7), according to GSEA comparison (Figure 7A, Appendix A, PLEKHO1 adjusted *p* = 0.00119, BGP4 adjusted *p* = 0.00728, and CCR7 adjusted *p* = 0.01041). In contrast, among the statistically significant, differentially expressed M2 genes, excluding allosome genes, Fc Fragment of IgG Receptor IIc (FCGR2C) was more highly expressed in females (Figure 7B, Appendix A, FCGR2C adjusted *p* = 0.01493) while Chromodomain Helicase DNA Binding Protein 9 (CHD9) was more highly expressed in men (Figure 7B, Appendix A, adjusted *p* = 0.02567). Among the allosome genes, significantly higher expression was measured for two X-linked genes in females and seven Y-linked genes in males (Appendix A).

Many immunomodulator genes were more highly expressed in mutant TP53 LUSC in females than males: C-X-C Motif Chemokine Ligand 9 (CXCL9), Major Histocompatibility Complex, Class II, DQ Beta 2 (HLA-DQB2), Major Histocompatibility Complex, Class II, DP Beta 1 (HLA-DPB1), T Cell Immunoreceptor With Ig and ITIM Domains (TIGIT), Major Histocompatibility Complex, Class II, DP Alpha 1 (HLA-DPA1), C-X-C Motif Chemokine Ligand 10 (CXCL10), Major Histocompatibility Complex, Class II, DQ Alpha 2 (HLA-DQA2), Major Histocompatibility Complex, Class II, DR Alpha (HLA-DRA), Indoleamine 2,3-Dioxygenase 1 (IDO-1), Programmed Cell Death 1 (PDCD1), Major Histocompatibility Complex, Class II, DQ Alpha 1 (HLA-DQA1), and CD40 Ligand (CD40LG) (Figure 7C, Appendix A, CXCL9 adjusted *p* = 0.00061, HLA-DQB2 adjusted *p* = 0.00358, HLA-DPB1 adjusted *p* = 0.00572, TIGIT adjusted *p* = 0.01007, HLA-DPA1 adjusted *p* = 0.02358, CXCL10 adjusted *p* = 0.02949, HLA-DQA2 adjusted *p* = 0.03444, HLA-DRA adjusted *p* = 0.03941, IDO-1 adjusted *p* = 0.04007, PDCD1 adjusted *p* = 0.04071, HLA-DQA1 adjusted *p* = 0.04095, and CD40LG adjusted *p* = 0.04714). CXCL9, TIGIT, IDO-1, and HLA-DQA1 are also part of the TIS and, together with Natural Killer Cell Granule Protein 7 (NKG7) and C-X-C Motif Chemokine Receptor 6 (CXCR6), are more highly expressed in females (Figure 7D, Appendix A, NKG7 adjusted *p* = 0.04740 and CXCR6 adjusted *p* = 0.04758). Two lymphocyte infiltration genes, CD3e Molecule (CD3E) and CD3d Molecule (CD3D), which encode protein components of the T cell receptor (TCR)-CD3 complex that is central to T cell activation and immune responses to foreign antigens [45] (Appendix A), were more highly expressed in mutant TP53 LUSC in females compared with their male counterparts (CD3E adjusted *p* = 0.05000 and CD3D adjusted *p* = 0.05588). Overall survival was not affected by relative gene expression of the significant immune signatures in mutant TP53 LUSC (Appendix A), marking a contrast to the instance of wt TP53 LUAD (Figure 2B–F). 

### 2.10. P53 Negative Regulators Are More Highly Expressed in wt TP53 LUAD 

In many cancer types, wt p53 protein levels and its function are held in check by elevated expression of its negative regulators, MDM2 and MDM4 (reviewed in [46]). To test whether this is the case in wt TP53 NSCLCs, the expression of MDM2 and MDM4 genes in both LUAD and LUSC was analyzed. Both MDM2 and MDM4 were expressed at higher levels in wt TP53 LUAD when compared to mutant TP53, without distinction between male and female cancers (Figure 8A *p* < 0.001, and Figure 8A,B *p* < 0.001, respectively). In contrast, both inhibitors were not differentially expressed between wt and mutant TP53 or between males and females for LUSC (data not shown).

### 2.11. PD-L1 Is More Highly Expressed in Mutant TP53 LUAD and Correlates with the Poorest Survival in wt TP53 LUSC

Given the reported correlation between mutant TP53 and PD-L1 (CD274) expression in LUAD [22], we tested this also as a function of patient sex in both LUAD and LUSC. In LUAD patients, PD-L1 expression levels were higher in the context of mutant TP53 than wt TP53 (Figure 8C), but this did not prove to be significantly different across the sexes (data not shown). Consistently, overall survival proved poorest for LUAD patients with the highest PD-L1 expression in conjunction with TP53 mutation (Figure 8D). In LUSC, PD-1 was not differentially expressed between wt or mutant TP53 cancers (data not shown). It was quite unexpected then that, in LUSC, coincidence between wt TP53 and highest PD-L1 expression levels corresponded with the poorest survival (Figure 8E).

Expression of the ligands of cytotoxic T-Lymphocyte Associated Protein 4 (CTLA-4): CD80 molecule (CD80) and CD86 molecule (CD86) were not significantly altered between wt and mutant TP53 LUAD (data not shown); while, in contrast, in LUSC they were significantly higher in mutant *TP53* compared with wt TP53 (Figure 9A, CD80 *p* = 0.0025 and Figure 9B, CD86 *p* = 0.00057).

## 3. Discussion

Our data revealed that the survival advantage of females with NSCLC (e.g., [12]) is not equivalent between its two main subtypes: LUAD and LUSC. Female patients with LUAD significantly outlive their male counterparts (Figure 1A,B), in contrast to the similar lifespan between the sexes for LUSC (Figure 5A,B). Consistent with the overall trend in nonreproductive cancers [17], survival is also greatest for LUAD when TP53 is wild-type (Figure 1C). Combining these two prognostic elements revealed that the clearest survival advantage in LUAD is for females with wt TP53 (Figure 1D,E).

Cancer patient outcomes are being increasingly linked to the immune response (reviewed in [47]). A direct correlation between tumor lymphocyte infiltration and cancer survival is being reported for lung and a number of other solid cancers [48]. It is notable that our study also identified an overall survival advantage for LUAD patients with lymphocyte-infiltration signature gene expression, while, on the other hand, worse survival for LUAD patients was observed to correspond with higher expression of genes in pathways of wound healing and TGF-β (Appendix A).

Sex-specific differences in immunity is indicated by the higher cancer incidence in males and the more frequent occurrence of autoimmune disease in females (reviewed in [49]). Our study in LUAD ties higher levels of tumor immune infiltrate to female patients with wt TP53 status (Figure 2A–D), which is the longest surviving group. In wt TP53 LUAD in females, the dominant immune signatures that we identified were INF-γ, lymphocyte infiltration, and enrichment of the M1 macrophage population (Figure 2E and Figure 3A, respectively), from among the categories defined by Thorsson et al., 2018 [33]. These pathways have clear roles in cancer defense, including: The role of INF-γ in tumor clearance [50], beneficial lymphocyte infiltration [48], and the pro-inflammatory and anti-tumor effects associated with M1 macrophages [51]. Intriguingly, inherent sex disparity in immunity between male and female mice has recently been attributed to differences in macrophage populations existing in advance of stimulation, which arise from three different tissues (peritoneal cavity, spleen, and Central Nervous System [CNS]). Genes of innate immunity pathways were found to mediate this disparity in the macrophages and noted to be stimulated by interferon [52].

The most dominantly expressed immune genes among wt TP53 LUAD in females mapped to antigen processing and presentation pathways feeding to Major Histocompatibility Complex I (MHCI) and MHCII (Figure 4A). The relevance of these two systems to empower the immune response against cancer is becoming more widely recognized [53]. These findings predict that the survival advantage among females with wt TP53 LUAD results from a superior capacity to combat cancer through CD8+ T cells and Natural Killer (NK) cell mediated-killing, directed through MHCI and also by CD4+ T cells through MHCII. 

LUSC, in contrast to LUAD, does not show a clear survival benefit in females in the US population (Figure 5A, Appendix A). While the overall frequency in males has declined since 2000, rates have remained fairly steady among females, as evident from US SEER data [5] (Appendix A). This reduction appears to reflect the decreased rate of males smoking unfiltered cigarettes [10], where smoke carcinogens are a major cause of the high levels of *TP53* mutations in this disease [18]. Unexpectedly, the small population of ~13% LUSC with wt *TP53* have poorer survival than those with mutant *TP53* (Figure 5B). Again, it cannot be presumed that TCGA data faithfully reflect the US population incidence, but, nonetheless, it is likely to reflect trends. This questions whether wt TP53 LUSC has distinct etiology and should be considered as an atypical LUSC subtype? On this point it is noteworthy that four of the cases considered atypical in a previous study ([34] and references within) were among this group.

Among the mutant TP53 LUSC group, female survival is longer (Figure 5D) and immune infiltrate is greater (Figure 6D) than for their male counterparts. In remarkable contrast to the findings in LUAD, it is the mutant TP53 LUSC females who have the advantageous immune signatures of IFN-γ, lymphocyte infiltration, and TGF-β, together with M1 macrophage signature (Figure 6E,F, respectively). Distinct immune genes were also discriminated between wt TP53 LUSC in males and females, but these were not found to predict survival (Figure 7). This data corroborated the evidence that LUAD and LUSC are clearly distinct diseases, predicting that therapeutic considerations should be made accordingly.

Regarding important therapeutic opportunities in NSCLC, for wt TP53 LUAD, the elevated expression levels of MDM2 and MDM4 (Figure 8A,B) in these cancers predict that these patients are likely to benefit from treatment with dual inhibitors of the MDM family (e.g., Aileron [54]). Mutant TP53 LUAD, on the other hand, identified to correlate with the poorest survival (Figure 1C) and was noted to have elevated PD-L1 expression levels. In fact, the poorest survival for mutant TP53 LUAD corresponded with the highest PD-L1 expression levels among these patients (Figure 8D). These findings predict that checkpoint inhibitors may have beneficial therapeutic application among this group of patients in particular. This finding is consistent with and significantly extends earlier work. A previous study [22] reported correlation between the expression of mutant TP53 and PD-L1 in LUAD. Survival analyses undertaken in the same study identified that the poorest survival corresponded with high levels of PD-L1 expression and low levels of TP53 expression. Without stratifying for TP53 status in these published survival analyses, however, conclusions were limited, particularly as p53 is largely regulated at the protein level. In our studies, by correlating poor survival to TP53 mutation in patients with high PD-L1 expression, we defined a group predicted to benefit most advantageously from PD-L1 checkpoint inhibitor.

The unexpectedly poorest survival in wt TP53 LUSC with higher PD-L1 expression (Figure 8C) and also elevated expression levels of CTLA-4 ligands CD80 and CD86 (Figure 9A,B, respectively) predicts benefit from checkpoint inhibitor therapy. While numerous genes have been correlated to *PD-L1* expression in lung [55], it is unclear what is driving *PD-L1* levels in wt *TP53* LUSC. However, it must be different from the reported mutant TP53/mir-34/PD-L1 axis in LUAD [22]. 

## 4. Materials and Methods 

### 4.1. Datasets

Australian dataset: Whole population-based data sourced from the Cancer Strategy & Development, Victorian Department of Health and Human Services, linked dataset, VCR 2013, 2017. Cox proportional hazard analyses for overall survival differences were adjusted for age and metastatic disease. To exclude co-mutation, such as Epidermal Growth Factor Receptor (EGFR), as a confounding prognosticator in LUAD, a database of surgical resections with minimum 22-gene next generation sequencing (NGS) panel was queried. The rates of EGFR and KRAS Proto-Oncogene, GTPase (KRAS) mutations and ALK Receptor Tyrosine Kinase (ALK) rearrangements were very similar, regardless of whether TP53 was co-mutated or wild type (data not shown).

SEER data: National Cancer Institute’s Surveillance, Epidemiology, and End Results (SEER) program [56]. The log-rank *p*-value for overall survival differences was adjusted for age, race, and tumor grade as confounders.

The Cancer Genome Atlas (TCGA) clinical data: TCGA LUAD and LUSC patient metadata data (including survival data) was obtained from The Broad Institute Genome Data Analyses Centre (GDAC) Firehose [57]. This clinical data was used to adjust for patient sex, age at diagnosis, tumor stage, race and smoking status for LUAD and also for LUSC, with the exception that smoking status was excluded (as reasoned in Section 2.7, that the 96% smoking history of LUSC predicted cause).

TP53 mutation status dataset: A full list of exome-level somatic mutations for TCGA LUAD and LUSC samples was downloaded from Genomics Data Commons Data Portal and GDAC Firehose [57]. Annotated mutations were analyzed using the Oncotator [58] cancer variant annotation tool. Mutations classified as “Silent”, ”Intron”, ”IGR”, or "lncRNA" were filtered out to focus on mutations with the potential of having a more profound functional impact. For the TP53 gene, the mutation status of each patient sample was classified as either wildtype-TP53 (wt TP53) or mutant TP53. 

TCGA expression data: Raw RNA-sequencing (RNA-Seq) gene level-counts for LUAD and LUSC were downloaded from The Broad Institute GDAC Firehose [57] and quantified using the software package RNA-seq by expectation-maximization (RSEM). Counts were converted to counts per million (CPM) and lowly expressed genes (<5 CPM were removed. CPM counts were normalized by library size using the R package edgeR [59].

Immune signature gene sets: Five immune signature gene sets were utilized to characterize the composition of immune infiltrate. These five immune signatures were identified by Thorsson et al., 2018 [33], to create a comprehensive and robust immune profile classification of more than 10,000 tumors across 33 cancer types in TCGA. These gene signatures were also associated to patient survival prognosis by cancer type. The five immune signatures are referenced below and the gene sets can be found in Appendix A.
Interferon-gamma response (IFN-γ) [60];Lymphocytes infiltration [35];Macrophage regulation [36];TGF-beta response (TGF-β) [37];Wound healing [38];

M1 and M2 macrophages gene signatures: The M1 and M2 gene signature was sourced from Martinez et al. 2006 [41]. The gene signatures can be found in Appendix A. 

Immunomodulators’ gene signature: The immunomodulator gene set was sourced from Thorsson et al. 2018 [33]. The gene list was curated from a literature review performed by immune-oncology experts within the TCGA immune response working group. The list of 78 immune-related genes represents agonist and antagonist genes that are either currently recognized or being evaluated for their utility in clinical oncology. The gene signature can be found in Appendix A.

Tumor inflammation signature (TIS): The TIS was shown to enrich for patients who respond to the anti-PD-1 agent pembrolizumab that is used in immunotherapy [42]. The signature, therefore, presents itself as a possible therapeutic marker of response to immunotherapy. The 18 gene signatures can be found in Appendix A.

### 4.2. Tumor Purity Calculation 

The tumor purity calculation for TCGA sample data was based on the ‘consensus measurement of purity estimation’ (CPE) method [30]. The CPE is the median purity estimate from the following methods: ABSOLUTE (a somatic copy-number measure), ESTIMATE (based on expression of a 141 immune gene set and a 141 stromal gene set [61]), LUMP (leukocytes unmethylation for purity that utilizes an average of 44 nonmethylated immune-specific CpG sites), and cell staining (discrimination according to hematoxylin and eosin sample staining). The tumor purity data can be found in Appendix A.

### 4.3. Immune Infiltrate Estimation

Immune infiltrate of TCGA sample data is an estimate of ‘leukocytes’ fraction’ based on DNA methylation arrays [33]. The ‘leukocytes’ fraction’ data can be found in Appendix A.

### 4.4. Propensity Score Adjustment Algorithm

Based on the propensity score algorithm developed by Rosenbaum and Rubin [62] and adapted by Yuan et al. 2016 [29], we adjusted for the following four TCGA clinical data fields as confounders in analysis between female and male LUAD cohorts: Age at diagnosis, tumor stage, race, and smoking history. For LUSC, only the confounding factors of age at diagnosis, tumor stage, and race were adjusted (as smoking was considered to be a cause, as explained in Section 2.7). Propensity scores based on patient sex were calculated using logistic regression. Then, based on these scores, a matching weight scheme was performed to re-weight the samples. This scheme balanced the propensity scores and ultimately the covariates. A strict checking loop was implemented following these steps so the propensity scores’ model was continuously revised until all covariates were balanced between males and females. Using the weights addresses skews in cofounders between female and male cohorts that may affect the interpretation of outputs (e.g., male cohorts may be overweight smokers relative to female cohorts). The weights calculated by the propensity scores’ algorithm were ingested by a number of statistical models to compare and assess significance for the different molecular data, including differential expression and gene set enrichment analysis.

### 4.5. Differential Expression Analysis

Using TCGA RNA-Seq normalized CPM, multiple groups were compared for differential gene expression: mutant TP53 tumors vs. wt TP53 tumors, female wt TP53 tumors vs. male wt TP53 tumors, and female Mt TP53 tumors vs. male mutant TP53 tumors. Differences in gene expression between these groups were computed with the R package limma (v3.41.17) [63] with adjustments made for confounders using the Propensity Scores (see Method Section 4.4). A Benjamini–Hochberg (BH) adjusted p-value was used to adjust for multiple hypothesis testing. Significant genes are inferred, using a 0.05 significance level.

### 4.6. Gene Set Enrichment Analysis (GSEA)

Gene set enrichment analysis (GSEA) [64] was undertaken to assess differentially expressed genes for enrichment of signatures of interest. GSEA was applied using the R package fgsea (v1.12.0). The rank metric used was logFold Change * (1-BH adjusted *p*-value). BH adjusted *p*-values from GSEA were derived. Significant gene sets were inferred, using a 0.05 significance level.

### 4.7. Survival Analysis 

Overall survival was assessed for TCGA LUAD and LUSC cohorts. Patients were stratified based on sex, TP53 mutation status, immune infiltrate levels, and gene expression level. Kaplan–Meier survival curves were used to show the survival differences between cohorts of patients. For cohorts based on immune infiltrate levels, immune gene signatures, or gene expression levels; the median value of immune infiltrate, gene set expression (the average normalized CPM of genes within the gene set), or the gene’s normalized CPM (respectively) was calculated to stratify individual patients into a high or low group. To assess differences in overall survival, a Cox regression model was used; including the covariates age, race, tumor stage, gender, and status for LUAD; and age, race, tumor stage, and gender for LUSC. The *p-*values were calculated using the log-rank test. Analyses were performed in R using survival (v2.44-1.1) and survminer (v0.4.6) packages.

### 4.8. Pathway Diagrams

Kyoto Encyclopedia of Genes and Genomes (KEGG) pathway diagram was generated in R using the pathview (v1.26.0) R package.

## 5. Conclusions

In summary, this in-depth bioinformatics’ analysis exposed the prognostic value of patient sex, TP53 status, and immune signatures in LUAD. It revealed TP53 status as an outstanding discriminator of survival in LUSC. Rational drug targets are predicted among subpopulations of lung cancers in this study. Together, these findings offer novel insights toward personalizing treatments against these highly aggressive human lung cancer subtypes. 

## Figures and Tables

**Figure 1 cancers-12-01535-f001:**
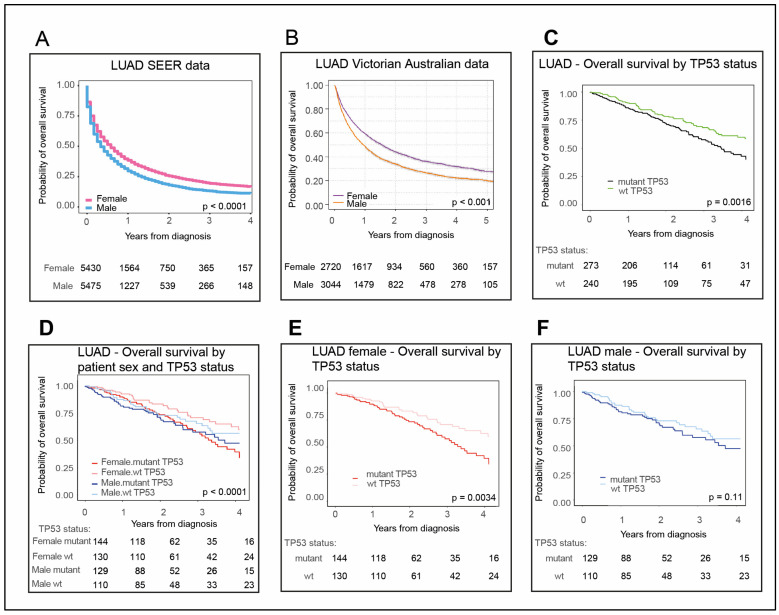
Females with lung adenocarcinoma (LUAD) and wild type (wt) TP53 outlive other LUAD patients. Overall survival data for LUAD (**A**) male (blue) and female (pink) patients from US Surveillance, Epidemiology, and End Results Program (SEER) data (2012–2016) and (**B**) male (orange) and female (purple) patients from Australian Victorian data (2013–2017). LUAD patient overall survival in The Cancer Genome Atlas (TCGA), (**C**) according to either wt TP53 (green) or mutant TP53 status (black); then for (**D**) females stratified for wt TP53 (pink) and mutant p53 status (red) and males stratified for wt TP53 (light blue) and mutant TP53 status (dark blue). Also shown separately for (**E**) women and (**F**) men. The numbers of patients corresponding to each cohort are tabulated below each plot. *‘p*’ is the p-value for differences in survival time between cohorts (significant *p*-value < 0.05). Refer to Section 4. Materials and Methods for description of methodology and datasets utilized in survival analysis.

**Figure 2 cancers-12-01535-f002:**
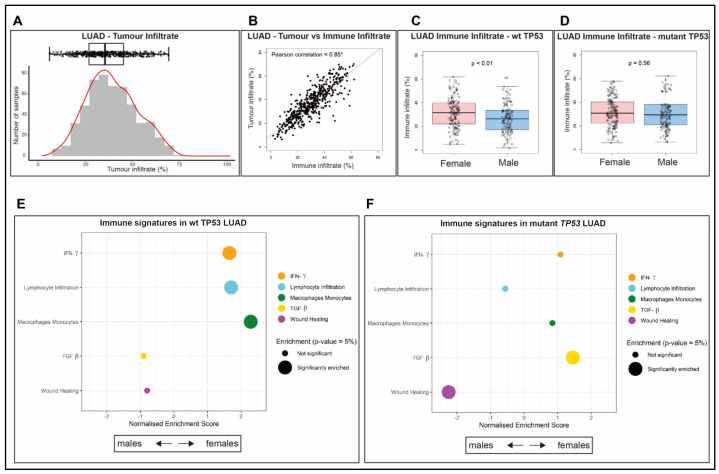
Female LUAD is highly immune infiltrated and enriched for specific immune signatures. (**A**) Proportion of tumor infiltrate in each TCGA LUAD sample. (**B**) Correlation between the proportion of tumor infiltrate and immune infiltrate in these samples. A Pearson correlation coefficient with the symbol * indicates the correlation is significantly different from 0, using the Spearman’s test of significance *(p-*value < 0.05). Proportion of immune infiltrate for females compared to males in LUAD samples with either (**C**) wt TP53*, p*-value < 0.01, or (**D**) mutant TP53, *p*-value = 0.56, using analysis of variance (ANOVA) to measure significance (*p-*value < 0.05). Gene Set Enrichment Analyses (GSEA) for immune signatures analyzing differentially expressed genes in LUAD females relative to males with (**E**) wt TP53 or (**F**) mutant TP53. Significantly enriched immune signatures (*p*-value < 0.05) are denoted by a large dot.

**Figure 3 cancers-12-01535-f003:**
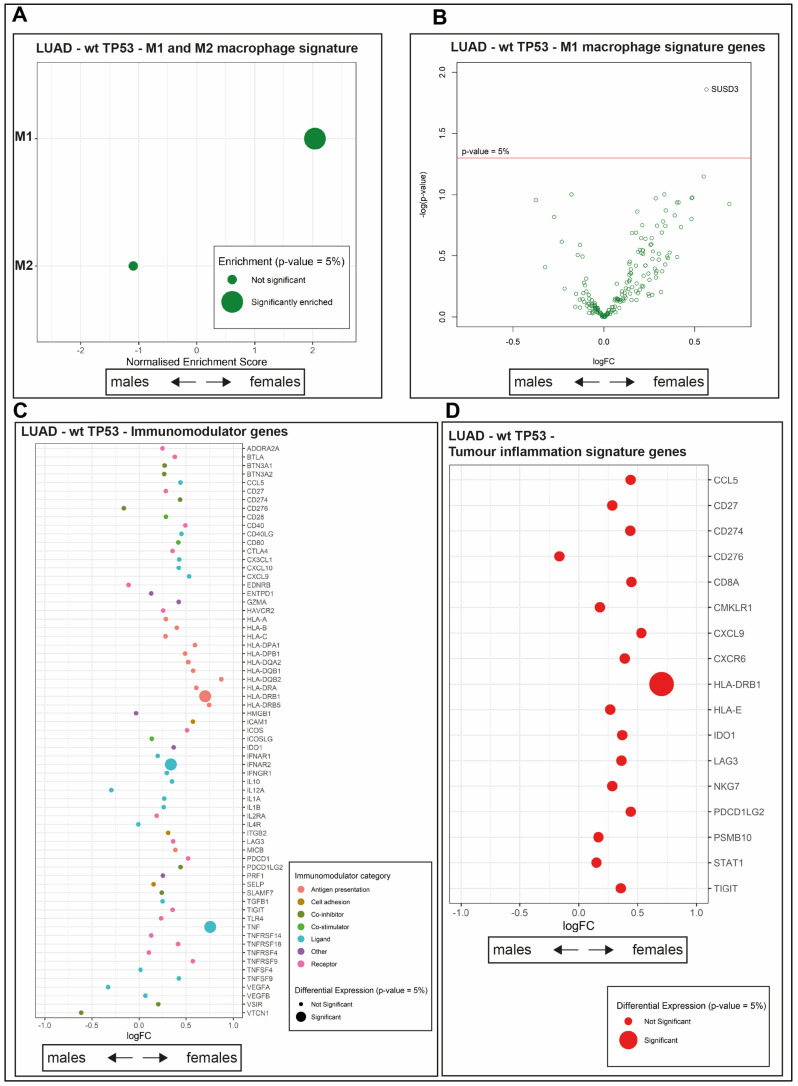
Sex-disparity in expression of immune genes in wt TP53 LUAD. Comparison of immune infiltrate genes expression between LUAD wt TP53 females and LUAD wt males using (**A**) GSEA of M1 and M2 macrophage genes, (**B**) differential gene expression analysis of M1 macrophage genes (with gene name assignment for *p*-values < 0.05), (**C**) differential gene expression of immunomodulator signatures, and (**D**) differential gene expression of tumor inflammation signature (TIS) genes. For panels **A**, **C** and **D**; significantly enriched immune signatures/genes *(p-*value < 0.05) are denoted by a large dot, respectively. A positive log fold change (logFC) demarks higher differential gene expression in females, while negative values indicate higher levels in males.

**Figure 4 cancers-12-01535-f004:**
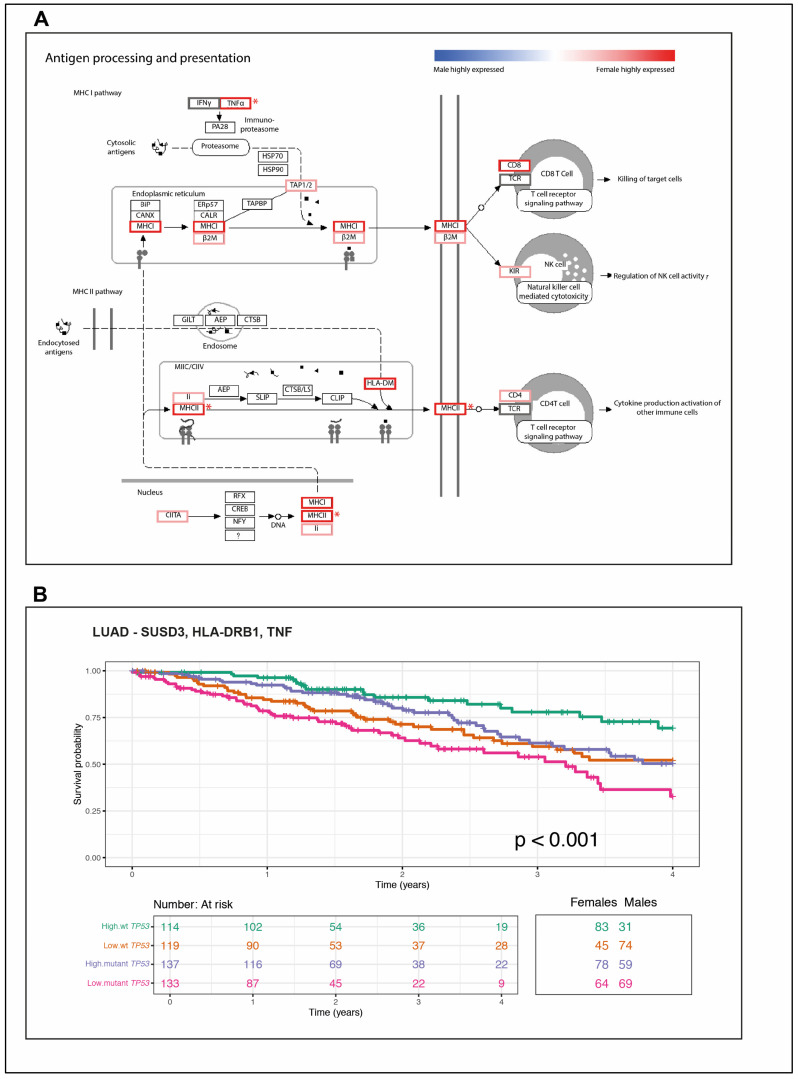
The wt *TP53* LUAD females are enriched in antigen processing and presentation pathways with links to longevity. Numerous immune signature genes that were identified to be most highly differentially expressed in wt TP53 LUAD in females, compared to their counterpart males, occur in the Kyoto Encyclopedia of Genes and Genomes (KEGG) antigen processing and presentation pathways as depicted (**A**). Note in this pathway diagram, Major Histocompatibility Complex Class II DR Beta 1 (HLA-DRB1) is referred to as Major Histocompatibility Complex, Class II (MHCII). Pathway genes with a positive log fold change (logFC) in females are outlined in red, while those with a negative logFC would be shaded blue (i.e., those more highly expressed in males). Grey outline indicates a logFC close to zero (i.e., similar expression levels between females and males). Genes with significant differential expression (*p-*value < 0.05) are indicated with a red asterisk. (**B**) Survival analyses for LUAD patients stratified by TP5*3* mutation status (wt or mutant) and the relative expression levels of the three gene signature: Sushi Domain Containing 3 (SUSD3), Major Histocompatibility Complex Class II DR Beta 1 (HLA-DRB1), and Tumour Necrosis Factor (TNF). The numbers of males and females corresponding to each cohort, stratified for high or low expression of the respective genes, in either a wt or mutant TP53 context are tabulated below each plot. The ‘*p*’ is the *p*-value for the log-rank test of differences in survival time (significant *p*-value < 0.05).

**Figure 5 cancers-12-01535-f005:**
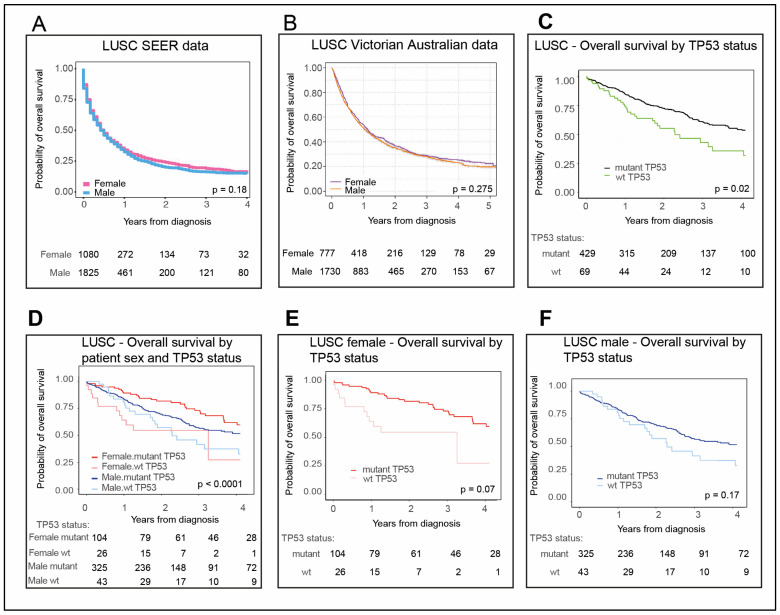
LUSC patient survival is longer for patients with mutant TP53 than wt TP53. LUSC overall survival data for (**A**) male (blue) and female (pink) patients in the US SEER datasets. (**B**) Male (orange) and female (purple) patients from Australian Victorian data. (**C**) LUSC patient overall survival in TCGA according to either wt p53 (green) or mutant p53 status (black), then for (**D**) females stratified for wt p53 (pink) and mutant p53 status (red) and males stratified for wt p53 (light blue) and mutant p53 status (dark blue). Also shown separately for (**E**) females and (**F**) males. The numbers of patients corresponding to each cohort are tabulated below each plot. The ‘*p*’ is the *p*-value for the log-rank test of differences in survival time between cohorts (significant *p-*value < 0.05). Refer to Section 4. Materials and Methods for description of methodology and datasets utilized in survival analysis.

**Figure 6 cancers-12-01535-f006:**
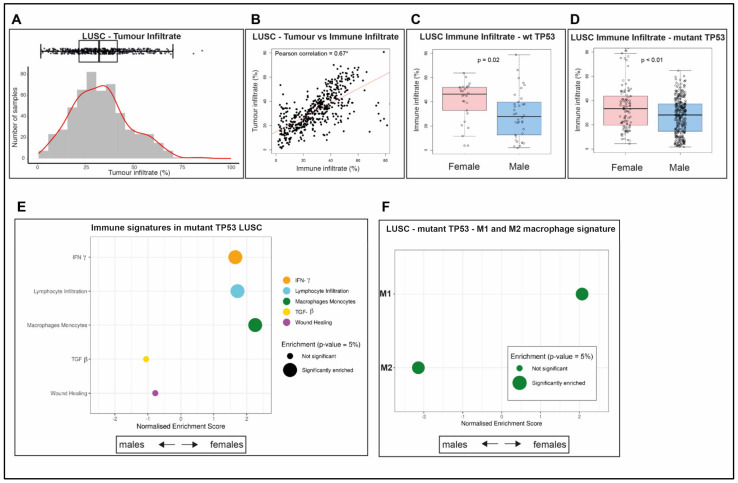
LUSC tumor infiltrate has a distinct immune content from LUAD. (**A**) Proportion of tumor infiltrate in each TCGA LUSC sample. (**B**) Correlation between the proportion of tumor infiltrate and immune infiltrate in these samples. A Pearson correlation coefficient with the symbol * indicates the correlation is significantly different from 0, using the Spearman’s test of significance (*p*-value < 0.05). Proportion of immune infiltrate for females compared with males in LUSC samples with either (**C**) wt TP53*, p-*value = 0.02, or (**D**) mutant TP53*, p-*value < 0.01, using analysis of variance (ANOVA) to measure significance (*p-*value < 0.05). GSEA of differentially expressed genes in LUSC females relative to males with mutant TP53 for **(E)** immune signatures **(F)** M1 and M2 macrophage signatures. Significantly enriched immune signatures (*p-*value < 0.05) are denoted by a large dot.

**Figure 7 cancers-12-01535-f007:**
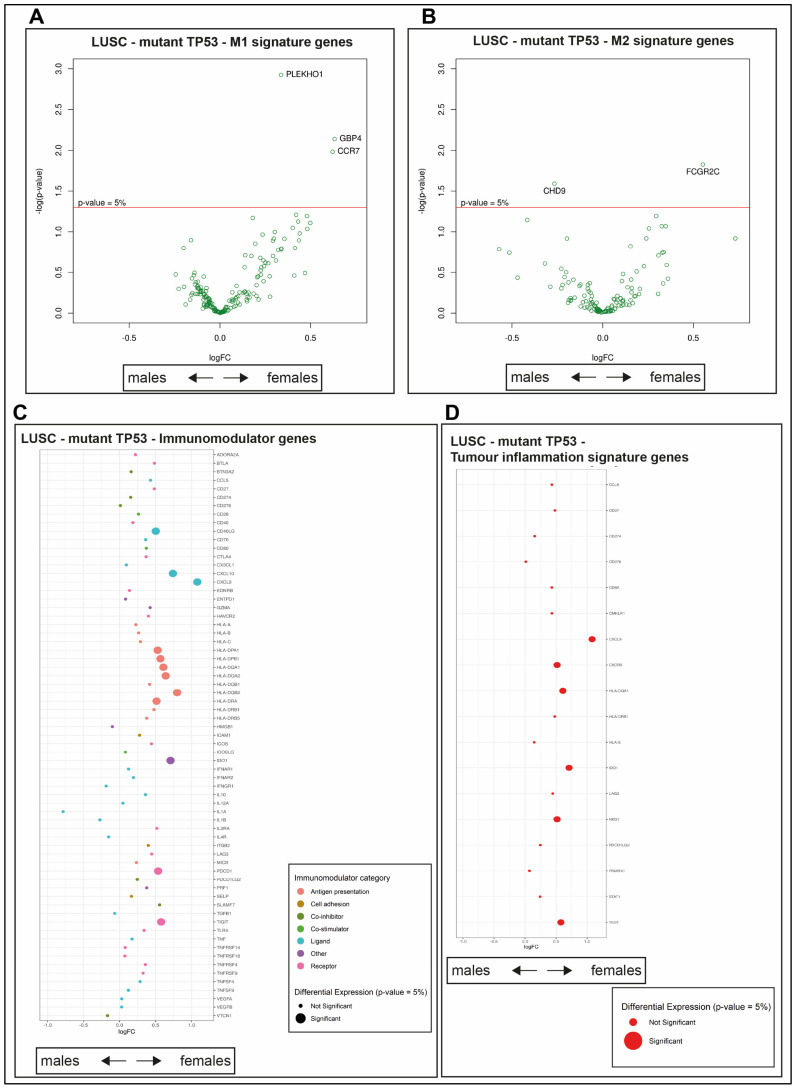
Sex disparity in expression of immune genes in mutant TP53 LUSC. Differential expression analysis of **(A)** M1 and **(B)** M2 macrophage genes (with gene name assignment for *p*-values < 0.05) **(C)** differential expression of immunomodulator gene signature genes, and (**D**) differential expression of tumor inflammation signature (TIS) genes. Significantly enriched genes (*p*-value < 0.05) are denoted by a large dot. A positive log fold change (logFC) demarks higher differential gene expression in females, while negative values indicate higher levels in males.

**Figure 8 cancers-12-01535-f008:**
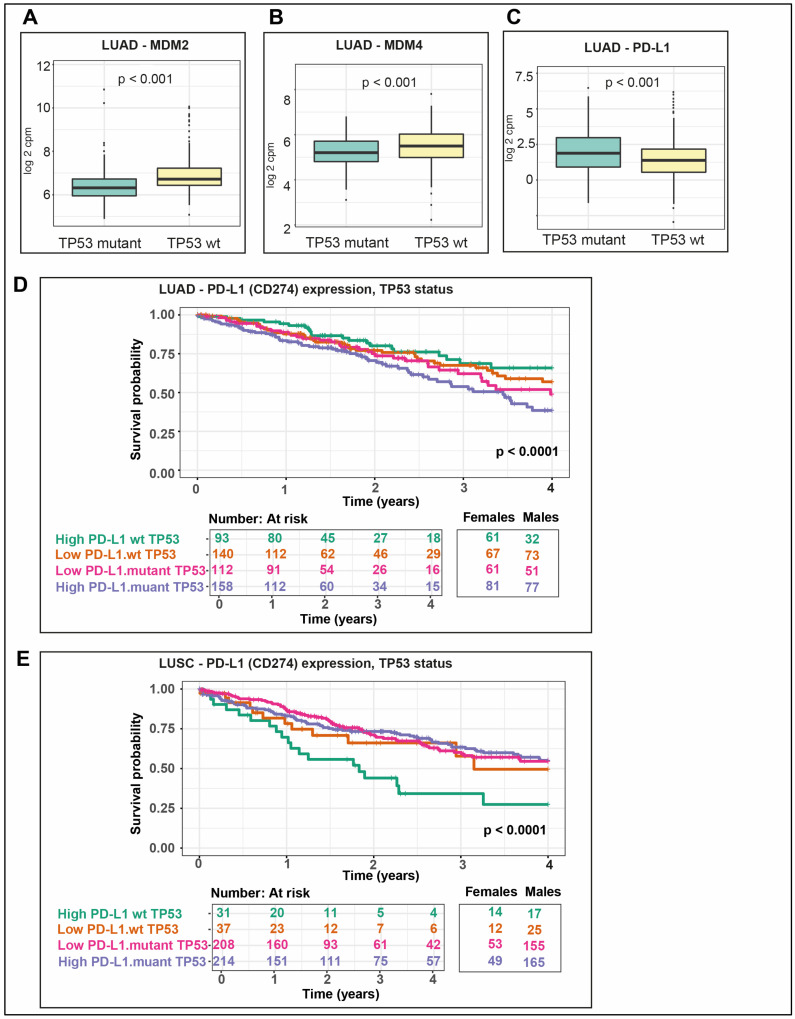
TP53 regulators and immune gene targets are expressed distinctly in LUAD and LUSC. In LUAD with wt TP53, expression is higher for its negative regulators (**A**) MDM2 and (**B)** MDM4, compared to those with mutant TP53. (**C**) PD-L1 expression is higher in LUAD mutant TP53 compared with wt TP53. Propensity scores were included and significance is *p* < 0.05. Survival differed between patients according to TP53 mutation status and PD-L1 expression levels in (**D**) LUAD and (**E**) LUSC.

**Figure 9 cancers-12-01535-f009:**
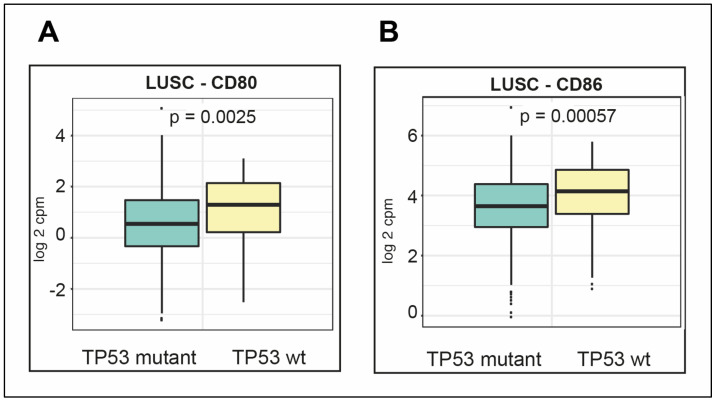
Expression of checkpoint Cytotoxic T-Lymphocyte Associated Protein 4 (CTLA-4) ligands: CD80 and CD86; differs in LUSC according to TP53 status. Expression is plotted for (**A**) CD80 and (**B**) CD86, for LUSC with either wt TP53 or mutant TP53. Propensity scores were included and expression differences assessed for significance (*p* < 0.05).

**Table 1 cancers-12-01535-t001:** TCGA patient samples for lung adenocarcinoma (LUAD) and lung squamous cell carcinoma (LUSC) listed according to patient sex and TP53 mutation status.

	Wildtype-TP53		Mutant-TP53		
Cancer	Female	Male	Female	Male	Total
LUAD	144	110	130	129	513
LUSC	26	43	104	325	498

Note: Cancer patient numbers from the TCGA clinical dataset. Refer to Section 4. Materials and Methods for a description of the methodology to identify patients with TP53 mutation. Wildtype-TP53 denotes tumors with wt TP53. Mutant TP53 denotes tumors with a pathogenic TP53 mutation.

**Table 2 cancers-12-01535-t002:** Prognostic value of enriched immune signature expression for LUAD patients.

Immune Signatures	Overall	Female	Male
IFN-G	-	-	-
Lymphocytes infiltration	Extended	Extended	-
Macrophage-Monocytes	-	Extended-	-
TGF-β	Poorer	-	Poorer
Wound healing	Poorer	Poorer	Poorer

Note for Table 2: For Overall, Female and Male cohorts respectively, the median expression value for each gene signature was calculated and compared to each individual patient gene signature mean expression, to classify patients into either the ‘High’ or ‘Low’ expression cohorts, respectively. The *p*-value from the Cox model using the Wald test of differences in overall survival time between ‘High’ and ‘Low’ expression cohorts was calculated. Considering the relative survival between the ‘High’ and ‘Low’ cohorts, respectively, if the *p*-value < 0.05, the gene signature was categorized as having either an “Extended” (blue) or “Poorer” (red) prognostic impact on survival. Otherwise the gene signature was labelled ‘-’, as having no significant prognostic impact on survival.

**Table 3 cancers-12-01535-t003:** Lack of prognostic value of immune signature expression in LUSC patients.

Immune Signatures	Overall	Female	Male
IFN-G	-	-	-
Lymphocytes infiltration	-	-	-
Macrophage-Monocytes	-	-	-
TGF-β	-	-	-
Wound healing	-	-	-

Note for Table 2: For Overall, Female, and Male cohorts, respectively, the median expression value for each gene signature was calculated and compared to each individual patient gene signature mean expression, to classify patients into either the ‘High’ or ‘Low’ cohorts, respectively. The *p-*value for the log-rank test of differences in overall survival time between ‘High’ and ‘Low’ expression cohorts was calculated. Considering the relative survival between the ‘High’ and ‘Low’ cohorts respectively, if the *p-*value < 0.05, the gene signature was categorized as having either an “Extended” (blue) or “Poorer” (red) prognostic impact on survival. Otherwise, the gene signature was labeled ‘-’, as having no significant prognostic impact on survival.

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
