# Peer review of "TP53 Status, Patient Sex, and the Immune Response as Determinants of Lung Cancer Patient Survival"

_cancers, 2020, doi:10.3390/cancers12061535_

Round 1
Reviewer 1 Report
The manuscript submitted by Freudenstein and colleagues evaluates a major concern that is often overlooked in many of the non-reproductive cancers, namely how sex influences tumorigenesis. In particular this group evaluated the influence of sex in non-small cell lung cancer patients and correlations with particular gene signatures and immune response. They confirmed that female LUAD patients live longer than their male counterparts from data procured from previous studies conducted in the US and Australia. Interestingly, while TP53 status was an overall predictor of survival, when stratifying based on sex, the prediction was strong in women but was not a significant predictor in men. The authors then used sample purity to assess immune infiltrates in the samples from the TCGA curated data. Identifying a variable amount of infiltrate, they assessed the infiltrate as a prognostic in male and female LUAD patients. Their data suggest that female patients with WT TP53 have higher levels of infiltrates than males with WT TP53. There were no correlations in the mutant p53 cases. Immune signatures between female and male LUAD patients were evaluated and GSEA was conducted. Further examination determined that an M1 macrophage signature is enriched in WT TP53 LUAD females as well as additional GSEA and KEGG analysis that further defined pathways associated with WT TP53 females. Interesting a somewhat inverse observation was found for LUSC patients, where WT TP53 appeared to be a poor prognostic factor for both females and males, although sample size reduced the power needed for significance.
Overall the study is comprehensive and begins to define a network of genes/pathways that could be influencing outcome between he sexes. The authors do a beautiful job of using already curated data to address a highly relevant biological and clinically-applicable question. The study is well conducted and thorough. Without questions, this information is imperative as we move to precision medicine. A thorough understanding of the cellular and molecular changes between females and males and how those changes influence survival, and therapeutic response will be critical.
Based on the comprehensive science presented, the timelines of the information, and the well-written manuscript there are no major concerns with the manuscript in its present form.
Author Response
We thank the reviewer for their time reviewing this manuscript and for their very positive review of our study.
Reviewer 2 Report
The given MS correlates the WT and mutant TP53 status, gender, and immunomodulators in patients with survival outcomes. The study is well conceptualized. However, it can be improvised with the following minor suggestions:
1) The title of the MS can be modified as TP53 status, gender, and the immune response as determinants on the survival of lung cancer patients.
2) The sidewise representation of each data with WT TP53 and mutant TP53 can help to understand clearly the differences.
3) The number of the dataset in the mutant TP53 in figure5, for example, is highly variable than the WT dataset. This might be the cause of the non-significant changes in the survival analysis. This needs to be explained.
4) The patient dataset extracted from public databases, the other factors affecting the outcomes like differences in the ethnicity, smoking habits, lifestyle, environmental factors have not taken into consideration. This and other shortcomings of the study need to be discussed.
5) The extensive modification with English, especially typos is needed.
Author Response
We thank the reviewer for their time reviewing our manuscript, we provide a point-by-point response below:
1) Regarding the request to modify the title to swap sex to gender: 'Patient sex' defines males and females genetically/biologically; on the contrary, gender is how one 'identifies', these are not the same and we purposefully chose 'sex' as the pertinent definition.
(2) It is unclear what a 'sidewise' presentation of the data means? As this was not a concern to Reviewer 1, we ask that the data presentation remain in its current form.
(3) It is correct that the number of datasets are highly variable, however this reflects the population sampling in TCGA. TCGA does not claim representative population sampling, but it must to a degree reflect the patient sub-types. The fact that there are not equal populations for each group indicates that each group is not equally prevalent. We worked within these constraints and discuss this in the text. Using the example of Figure 5: as we discuss, smoking is the major cause of LUSC and is highly associated with TP53 mutation. Due to smoking prevalence, we considered it as a founding factor associated with TP53 mutation. In this context we undertook our subsequent immune analyses on the mutant TP53 population only.
(4) As stated the SEER data is total population sampling and has been adjusted for age, race and tumour stage as now indicated in the materials and methods section 4.1. The Australian data has now been adjusted for age and metastatic stage as also indicated in the section 4.1 (where race is not included as > 90% of the sampled population is of European decent). All the TCGA data were analysed considering confounding factors as stated in the methods.
(5) We have altered the text for consistent English spelling throughout (eg. tumour, analyse). For clarity and consistent with the type-set title, we have now defined the gene as TP53 and the protein as p53, and have corrected the text throughout accordingly. Other changes have been corrected.
We noted that Yuan et al reference was included in the references twice and corrected this and included our just published review (new reference 24).
We also noted that Figure 9 appeared to have been excluded from the manuscript, so this has now been added.